# FEDSRC: FEDERATED LEARNING WITH SELF-REGULATING CLIENTS

## ABSTRACT

Federated Learning (FL) has emerged as a prominent privacy-preserving decentralized paradigm for collaborative machine learning across many devices. However, FL suffers from performance degradation in the global model due to heterogeneity in clients' locally generated data. Some prior studies address this issue by limiting or even discarding certain clients' contributions to the global model, resulting in unnecessary computation and communication for the discarded clients. Alternatively, selectively choosing clients to participate in FL may avoid such resource waste. But, such active client selection requires client-level profiling that violates privacy. In this paper, we present a novel FL approach, called FedSRC: **Fed**erated Learning with **S**elf-**R**egulating **C**lients, that can save clients' resources while preserving their anonymity. In FedSRC, clients can determine themselves if their local training is favorable to the global model and whether they should participate in an FL round using a lightweight checkpoint based on a local inference loss on the global model. Through comprehensive evaluations using four datasets, we show that FedSRC can improve global model performance, all the while reducing communication costs by up to 30% and computation costs by 55%.

## 1 INTRODUCTION

**Motivation.** Federated Learning (FL) is a popular privacy-preserving distributed Machine Learning (ML) approach that has been implemented in widely used applications like Google's Gboard Hard et al. (2019) and Apple's Siri Granqvist et al. (2020). In FL, many clients collaboratively train a shared ML model through iterations where clients locally train the shared model using their private data and anonymously send back their updated model. FL enjoys the advantages of training with a larger dataset from many clients, yet clients' data never leaves their devices, offering enhanced privacy. A central FL server facilitates the iterations by aggregating the clients' model updates into the shared global model.

A well-documented drawback of FL's siloed decentralized training is the slow convergence and poor performance of the global model due to statistical differences (i.e., not independent or identically distributed (non-IID)) among the clients' data Li et al. (2020b). On top of the naturally occurring data variation among FL clients, another source of such statistical differences is the *data quality*. The quality and reliability of the locally collected client data by different hardware and sensors may vary significantly, especially in mobile and wearable devices Cho et al. (2021a;b). Device manufacturers utilize hardware/sensors of varying qualities to meet their own goals of device functionality and price points. Moreover, continued innovation in mobile devices is yielding increasingly high-quality data with newer generations of sensors and hardware Haghi et al. (2017); Cheng et al. (2021). A recent Facebook study identifies thousands of different types of hardware among the devices using their application Wu et al. (2019). In addition, malfunctioning devices can also be responsible for feeding FL with bad-quality data Liu et al. (2020). Worse yet, the FL clients' devices and sensors are also subject to malicious attacks that may intentionally corrupt client data to poison the global model, exacerbating FL's data quality issue Tolpegin et al. (2020).

**Limitations of existing approaches.** A prominent line of prior work aims at handling the aforementioned data heterogeneity/quality issue by controlling the contributions from different clients Li et al. (2022); Karimireddy et al. (2020); Yin et al. (2018); Talukder & Islam (2022). This approach is built upon the idea that the updates from certain clients (e.g., clients with bad data quality) are un-

favorable for the global model and should be given a lower weight in FL's centralized model update aggregation. While this can improve FL performance, a major limitation of the aggregation-weight-based approach is that clients whose model updates receive a lower weight and, hence, contribute little to the central model still go through the computationally hungry model training and communicate the updated model to the central FL server. This unnecessary use of clients' computation and communication resources makes FL training inefficient.

An alternative to the wasteful weight-based approach to handle FL's data heterogeneity is "active client selection" where the central server profiles the quality of clients' model updates and selects only "favorable" clients to participate in FL training Cho et al. (2022); Goetz et al. (2019). While active client selection does not waste resources, this approach needs to tag client updates with client IDs for the active selection. Therefore, it cannot maintain clients' anonymity and diminishes FL privacy.

**Our contribution.** We recognize that both the above-mentioned resource waste and breach of anonymity can be avoided (while still handling data quality issues) if the clients themselves can anticipate the resource waste and refrain from participating in model updates. To enable this client-side active client selection, we propose a novel FL approach where the clients actively regulate their own participation. We call this **Fed**erated Learning with **S**elf-**R**egulating **C**lients or FedSRC in short. In FedSRC, clients implement a "checkpoint" in its local training path to determine whether it should continue and finish training and send the model update back to the FL central server. A client saves the computation cost of local training and the communication cost of sending the model update if it decides to exit the FL round. On the other hand, client anonymity is not violated, as the central server does not need any client profiling for client selection. Furthermore, FedSRC's client-side implementation can still be paired with heavier centralized FL techniques that tackle model and data poisoning attacks at the central server. To the best of our knowledge, FedSRC offers the first variation of FL that allows clients to make strategic decisions to aid the FL global model.

However, implementing FedSRC's active client selection (i.e., participation checkpoint) that handles FL's data quality issue is challenging. In FL, clients only have access to their own data and, therefore, cannot statistically determine their data quality and employ strategic FL participation. Moreover, our selection strategy must be lightweight since it needs to be deployed on the client device.

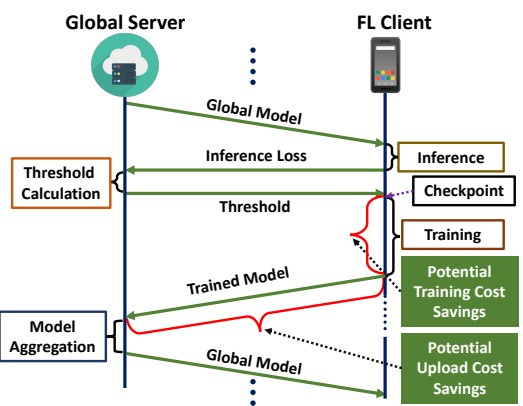

Figure 1: FedSRC adds one checkpoint based on inference loss during training for regulating client participation.

With these constraints in mind, we develop an inference loss-based participation policy where the clients utilize the global shared model as a "litmus test" for their data quality and exit the FL training if they have high local inference loss on the global model. Our design is motivated by our empirical observation that, in general, clients' low-quality data results in higher local inference loss on the global model. Fig. 1 illustrates the main working principle of FedSRC while the details are presented in Section 3.1.

We also offer the first-ever theoretical analysis of FL convergence under clients with data quality issues. Our analysis reveals that FedSRC's strategic selection can boost both the performance and convergence rate of FL. We evaluate FedSRC using four different datasets and show that with the same number of communication rounds, FedSRC can save as much as 30% on communication and 55% on computational cost.

## 2 PRELIMINARIES

### 2.1 FEDERATED LEARNING

**Problem formulation.** Suppose in a federated setup, there are $K$ clients, each with their own dataset $\mathcal{D}_k$. The objective of FL is to minimize the global loss $F(w)$, which can be expressed as

$$F(w) = \frac{1}{\sum_{k=1}^{K} |\mathcal{D}_k|} \sum_{k=1}^{K} \sum_{\xi \in \mathcal{D}_k} f(w, \xi) = \sum_{k=1}^{K} p_k F_k(w) \tag{1}$$

where $f(w, \xi)$ is the composite loss function for sample $\xi$ and model parameter $w$, $p_k = \frac{|\mathcal{D}_k|}{\sum_{k=1}^{K} |\mathcal{D}_k|}$ is the fraction of data at the $k$-th client, and $F_k(w) = \frac{1}{|\mathcal{D}_k|} \sum_{\xi \in \mathcal{D}_k} f(w, \xi)$ is the local loss function of client $k$.

**Solution.** The FedAVG McMahan et al. (2017) algorithm minimizes Eq. equation 1 efficiently by dividing training into multiple rounds. In each round $t$, a fraction $C$ of clients ($m = CK$) is randomly selected from $K$, and the selected clients are denoted by $S^{(t)}$. Selected clients perform $\tau$ local SGD iterations and update their models, which are then aggregated into a new global model. Accordingly, the model update for a client can be written as follows:

$$w_k^{(t+1)} = \begin{cases} w_k^{(t)} - \eta_t g_k(w_k^{(t)}, \xi_k^{(t)}) & \text{if } (t+1) \bmod \tau \neq 0 \\ \frac{1}{m} \sum_{l \in S^{(t)}} \left( w_l^{(t)} - \eta_t g_l(w_l^{(t)}, \xi_l^{(t)}) \right) = \bar{w}^{(t+1)} & \text{if } (t+1) \bmod \tau = 0 \end{cases}$$

where $w_k^{(t+1)}$ denotes the local model parameters of client $k$ at iteration $t$, $\bar{w}^{(t+1)}$ is the global model, $\eta_t$ is the learning rate, and $g_k(w_k^{(t)}, \xi_k^{(t)}) = \frac{1}{b} \sum_{\xi \in \xi_k^{(t)}} \nabla f(w_k^{(t)}, \xi)$ is the stochastic gradient over mini-batch $\xi_k^{(t)}$ of size $b$ which is randomly sampled from local dataset $\mathcal{D}_k$ of client $k$. The global model, $\bar{w}^{(t)}$, is only updated after every $\tau$ iteration. But, for the purpose of our analysis, we consider a virtual sequence of $\bar{w}^{(t)}$ that is updated at each iteration as follows:

$$\bar{w}^{(t+1)} = \bar{w}^{(t)} - \eta_t \bar{g}^{(t)} =: \bar{w}^{(t)} - \frac{\eta_t}{m} \sum_{k \in S^{(t)}} g_k(w_k^{(t)}, \xi_k^{(t)}).$$

### 2.2 IMPROVING FL WITH DATA QUALITY ISSUES

**Biased aggregation.** In FL, treating every client equally (e.g., model aggregation of FedAVG) when they have data quality issues may lead to severe performance degradation of the global model Talukder & Islam (2022). Unlike centralized training, FL suffers more from data quality issues. FL clients train their ML models locally, and their model updates can deteriorate significantly due to bad data, eventually affecting the aggregated (with equal weights) global model. To mitigate this, several biased FL aggregation policies have been developed Li et al. (2022); Karimireddy et al. (2020); Talukder & Islam (2022). However, while a biased aggregation improves global performance, it can also be seen as "unfair" to clients who get low or zero weights and, consequently, do not benefit from federation. After all, FL is intended for collaborative training across many participating clients. Nevertheless, we argue in favor of biased aggregation since clients with bad data quality suffer from worse model performance on their local data anyway; including them in model aggregation only harms the global model for everyone else.

**Self-regulating clients.** Implementing the biased aggregation in the central server is inefficient since it requires every client, even those with data quality issues, to complete the local training and model update. To avoid this wasteful (for clients with bad data) model training and updating the central server, we adopt *biased aggregation through client selection*, i.e., we select the clients with good data to participate in FL training. However, client selection in such a manner requires profiling of clients' data quality and, hence, if implemented at the central server, breaks clients' anonymity. Consequently, to maintain client anonymity even through the client selection process, we need the clients to be able to profile and apply the selection to themselves; in other words, clients need to self-regulate their FL participation.

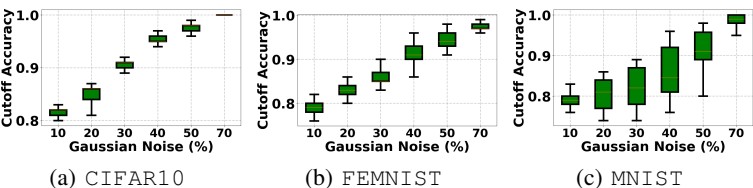

Figure 2: Histograms of clients' test losses on the global model of one training round of FedAVG for different dataset in presence of bad clients.

Figure 3: Accuracy of inference loss-based detection (cutoff) of good and bad clients across the training rounds.

**Challenges.** Implementing self-regulating clients with strategic client selection is non-trivial. *First*, clients only have access to their own data, which may not be enough to apply statistical methods to determine the quality of data effectively. *Second*, unlike the central server, the clients also do not have access to other clients' model updates. *Third*, the selection strategy must be lightweight with low overhead since FL clients are resource-constrained. In what follows, we develop a lightweight client selection policy to address these challenges.

## 3 OUR SOLUTION

### 3.1 FedSRC: FEDERATE LEARNING WITH SELF-REGULATING CLIENTS

**Client classification.** To implement a client selection strategy, we first need to define who should be considered as a "bad client" and discarded from model aggregation. Since FL clients' data is private, we cannot directly assess clients' data quality. Hence, we classify bad clients based on the impact of their inclusion in the global model as follows:

**Definition 1** ($\epsilon$-Bad Client). *An FL client is $\epsilon$-bad if its inclusion in the unbiased global aggregation increases the converged global objective loss by more than $\epsilon$.*

The parameter $\epsilon$ in our definition serves two purposes. *First*, it allows us to set the degree of negative impact that constitutes a bad client. *Second*, it can absorb the variation of global objective loss (for good client participation) due to non-IID data distribution and the sequence of client participation in the training rounds. Our definition, however, can only be an *approximate* definition of bad clients since we cannot distinguish the impact of bad data (from bad clients) and non-IID data (from good clients) on the global loss. Nevertheless, our definition serves to develop a client selection strategy for improving the global model performance, albeit there is a possibility (tunable through $\epsilon$) of treating some good clients with non-IID data as bad clients.

**Client selection strategy.** The client classification in Definition 1 requires $N + 1$ complete FL training for $N$ clients, rendering it impractical due to its huge computation and communication overheads. Moreover, this classification also breaks client anonymity at the central server. Hence, we need to develop a lightweight approach for identifying good and bad clients at the client level that can serve as a proxy for Defintion 1.

We devise FedSRC's client selection strategy based on our empirical observation that *clients with bad quality data suffer from worse performance when vetted against the global model*. More specifically, we find that the bad clients tend to have a higher inference loss on the shared global model across training rounds, even when they are included in the model aggregation. To demonstrate this, we run experiments on several data sets with 30% of clients suffering from noisy data (more details

---

**Algorithm 1** FedSRC

---

**Input:** Initial global model $(w_0)$

1: **for** each round $i = 0$ to $t$ **do**
2:     *Global model sharing (Server):* The central server randomly selects a subset of FL clients and sends them the latest global model.
3:     *Local inference (Client):* The clients run inference on the global model with a random subset of their training data set and send back the inference loss to the central server.
4:     *Setting participation threshold (Server):* The central server collects the test losses from the clients, determines the participation threshold, and then broadcasts the threshold to the participating clients.
5:     *Self-Regulating participation (Client):* The clients check their test loss against the server's threshold. A client stops training and drops from the FL round if its test loss exceeds the participation threshold.
6:     *Local training (Client):* Participating clients complete the training and send the updated model to the central server.
7:     *Model aggregation (Server):* Central server aggregates the model updates and prepares the global model for the next FL round.
8: **end for**

---

of data sets and how we add the noise can be found in Section 4.1 and Appendix B). Fig. 2 shows the histograms of clients' test losses on the global model of one training round of FedAVG for different data sets. We can clearly see that the test losses of the bad clients are distinguishably higher than those of good clients, and we can set a cutoff/threshold to separate the good and the bad clients. To investigate the efficacy of an inference loss threshold-based approach, we then vary the clients' data quality by changing the amount of noise in the bad clients' data. We track the accuracy of an inference loss-based detection of good and bad clients and show them in Fig. 3. We see that, even when there is only 10% Gaussian noise added to the bad clients, an interference loss threshold can identify the good and bad clients with ∼80% accuracy.

The inference on the client side is not computation-heavy and can be done on a randomly chosen subset of a client's training data. An inference loss-based approach satisfies our requirements for client-side regulation since it can be a reasonably accurate proxy for Definition 1. Consequently, we set our client selection strategy as follows: *during FL training, we select the clients with inference loss lower than a given threshold.*

**Threshold-based participation.** While we would like the clients to implement our selection strategy and set the participation threshold themselves, they do not have access to the inference losses of other clients. Hence, in FedSRC, we engage the central server to *anonymously* collect the inference losses of participating clients' of a particular FL round and determine the participation threshold. This is an additional step we introduce in FedSRC. We defer the discussion of the overhead associated with FedSRC to the end of this section.

**Setting the participation threshold.** Since we have the inference losses of both good and bad clients, the central server can set the threshold *autonomously* by running unsupervised clustering to break the inference losses into two groups and setting the cluster boundary as the threshold. A computationally lighter alternative to the autonomous approach above is utilizing insight into user data sets, such as the expected percentage of clients with bad data or user-defined participation policy, such as discarding a certain percentage of clients every round. The central server can then set the threshold accordingly to satisfy the externally determined participation percentage. Ideally, with perfect separation between good and bad clients, the user-supplied percentage should match the percentage of bad clients participating in the FL round. While determining the percentage of bad clients in a real-world scenario is non-trivial, we find in our evaluation that overestimating the percentage of bad clients is more favorable than underestimating (Fig. 11(a) in the Appendix C.3). The intuition behind this observation is that including a few bad clients is more harmful to the global model than missing the contribution from a few good clients.

After setting the threshold, the central server broadcasts the threshold loss to all clients selected for that specific training round. At no point in FedSRC does the central server need to track the source

of the data (i.e., client ID) it collects from clients. We summarize the implementation of FedSRC in Algorithm 1.

**Overhead of** FedSRC**'s implementation.** FedSRC's checkpoint adds minor computational overhead to the client as we add one additional inference on a subset of the client's training data. But the added inference cost is negligible compared to the training cost savings. We can also tap the initial minibatch error before the global weight is modified to estimate the inference loss of the clients when the minibatch is randomly sampled from the training data. In FedSRC, clients also have additional communication with the central server to send their test losses. However, the clients send only one value to the server, hence, the extra communication cost is negligible. Nevertheless, to collect the test losses from all clients reliably, the server may need to offer longer response deadlines, thereby leaving FL clients waiting for the participation threshold.

## 3.2 THEORETICAL ANALYSIS

Here, we prove the convergence of FedSRC and discuss how our client selection policy affects the convergence. To facilitate our analysis, We make the following assumptions:

**Assumption 1.** $F_1, \ldots, F_k$ are all L-smooth, i.e., for all $v$ and $w$,

$$F_k(v) \leq F_k(w) + (v - w)^T \nabla F_k(w) + \frac{L}{2}\|v - w\|_2^2.$$

**Assumption 2.** $F_1, \ldots, F_k$ are all $\mu$-strongly convex, i.e., for all $v$ and $w$,

$$F_k(v) \geq F_k(w) + (v - w)^T \nabla F_k(w) + \frac{\mu}{2}\|v - w\|_2^2.$$

**Assumption 3.** For the mini-batch $\xi_k$ uniformly sampled at random from $\mathcal{D}_k$ of user $k$, the resulting stochastic gradient is unbiased; that is, $\mathbb{E}[g_k(w_k, \xi_k)] = \nabla F_k(w_k)$. Also, the variance of stochastic gradients is bounded: $\mathbb{E}[\|g_k(w_k, \xi_k) - \nabla F_k(w_k)\|^2] \leq \sigma^2$ for all $k = 1, \ldots, K$.

**Assumption 4.** The stochastic gradients' expected squared norms are uniformly bounded, i.e., $\mathbb{E}[\|g_k(w_k, \xi_k)\|^2] \leq G^2$ for $k = 1, \ldots, K$.

Denote by $\mathcal{B}$ the set of $\epsilon$-bad clients for a fixed $\epsilon > 0$, and let $\mathcal{G}$ be the set of good clients (i.e., those that are not $\epsilon$-bad. By Definition 1, these sets are fixed.

Since our assumption is that there are bad clients whose updates adversely affect the global model, our convergence analysis takes this into account by separating the good and bad clients in all terms defined below. We utilize similar ideas to Cho et al. (2022) by defining a local-global objective gap and a skewness of biased selection of clients who send their model update to the central server. In contrast to prior work, our definitions are in terms of the good (or potentially bad) clients, which allows us to understand the effect of our client selection strategy in the context of our problem setup.

We define the global loss for two client sets: $F_g(w) = \sum_{k \in \mathcal{G}} p_k F_k(w)$ for the good clients in $\mathcal{G}$, and similarly define $F_b$ for the bad clients in $\mathcal{B}$. The optimal global losses for good and bad clients are $F_g^* = \min_w F_g(w)$ and $F_b^* = \min_w F_b(w)$. Additionally, we define the global model optimum $w^* = \arg\min_w F(w)$, and the client-level optima $w_k^* = \arg\min_w F_k(w)$ for each client $k$.

**Definition 2** (Local-Global Objective). *We define the local-global objective gap for the set of good clients as follows:*

$$\Gamma_g = F_g^* - \sum_{k \in \mathcal{G}} p_k F_k^* = \sum_{k \in \mathcal{G}} p_k(F_k(w^*) - F_k(w_k^*)) \geq 0. \tag{2}$$

For highly non-iid data, $\Gamma_g$ is non-zero, and larger $\Gamma_g$ implies higher data heterogeneity. $\Gamma_g = 0$ implies consistent optimum models among the clients and the central server.

**Definition 3** (Selection Skewness). *Let $w$ be the current weights of the global model, and $\pi$ be any client selection strategy. We let $S(\pi, w)$ denote the selected clients using selection strategy $\pi$ and define the skewness of the client selection strategy $\pi$ for good and bad clients via*

$$\rho_g(S(\pi, w), w') = \frac{\mathbb{E}_{S(\pi,w)}\left[\frac{1}{p}\sum_{k \in S(\pi,w) \cap \mathcal{G}}\left(F_k(w') - F_k^*\right)\right]}{F_g(w') - \sum_{k \in \mathcal{G}} p_k F_k^*}, \tag{3}$$

$$\rho_b(S(\pi,w),w') = \frac{\mathbb{E}_{S(\pi,w)}\Big[\frac{1}{q}\sum\limits_{k\in S(\pi,w)\cap \mathcal{B}}\big(F_k(w')-F_k^*\big)\Big]}{F_g(w')-\sum\limits_{k\in\mathcal{G}}p_k F_k^*},\tag{4}$$

*where p is the number of selected good clients, q is the number of selected bad clients, and $m=p+q$. Above, the current global model weights w influences the selection strategy $\pi$, while $w'$ is the global model weight at which the selection skewness is evaluated. $\mathbb{E}_{S(\pi,w)}[\cdot]$ represents the expectation over the randomness from the selection strategy $\pi$ in determining $S(\pi,w)$.*

Note that the denominator of both $\rho_g$ and $\rho_b$ are the same, and represent the current gap between the local and global models for good clients only. This is because we do not wish to select the bad clients, and their local-global objective gap should not influence our convergence analysis. The following terms are useful for providing a concrete error bound in the main theorem below.

$$\bar\rho_g = \min_{w,w'}\rho_g(S(\pi,w),w'),\tag{5}$$

$$\tilde\rho_g = \max_{w}\rho_g(S(\pi,w),w^*).\tag{6}$$

We define $\bar\rho_b$ and $\tilde\rho_b$ similarly.

**Theorem 1.** *Under the Assumptions stated above, for a learning rate $\eta_t=\frac{1}{\mu(t+\gamma)}$ with $\gamma=\frac{4L}{\mu}$, and for client selection strategy $\pi$ that selects the same number of good and bad clients (p and q, respectively) after time T, the error of federated learning with self-regulating clients satisfies, for every $t\ge T$,*

$$\mathbb{E}[F(\bar w^{(t)})]-F^* \le \underbrace{\frac{1}{(t+\gamma)}\left[\frac{4L(32\tau^2 G^2+\sigma^2/m)}{3\mu^2\bar\rho_g}+\frac{8L^2\Gamma_g}{\mu^2}\frac{\bar\rho_b}{\bar\rho_g}+\frac{L(\gamma+1)(\|\bar w^{(1)}-w^*\|^2)}{2}\right]}_{\text{Vanishing Term}}$$
$$+\underbrace{\frac{8L\Gamma_g}{3\mu}\left(\frac{p\tilde\rho_g+q\tilde\rho_b}{m\bar\rho_g}-1\right)}_{\text{Bias Term}}$$

$$\tag{7}$$

To the best of our knowledge, Theorem 1 provides the first theoretical bound of convergence for federated averaging in the presence of bad clients. The complete proof of the theorem can be found in Appendix A.

**Effect of the client selection strategy.** First, note that for an unbiased client selection strategy (clients participate in the model update uniformly at random), both good and bad clients will provide a model update. As the training of the model progresses, the loss of the good clients decreases, whereas the loss of the bad clients does not improve. This results in a decreasing $\rho_g$, but increasing $\rho_b$, both of which negatively affect both the rate of convergence of the vanishing term and the magnitude of the bias term in equation 7. A biased client selection strategy that is able to discard clients with higher loss will ensure an increase in the number of good clients selected and decrease in number of bad clients selected, which reduces the value of $\rho_b$ and increases the value of $\rho_g$, resulting in both faster convergence and smaller bias.

**Reducing $\rho_b$ and increasing $\rho_g$ for faster convergence.** Under our model for good and bad clients, if our selection strategy prioritizes client updates for those with small testing loss value $F_k$, the number of bad clients selected in $S(\pi,w)$ will be smaller, which results in larger $\rho_g$ but smaller $\rho_b$. Consequently, the first two terms in the vanishing term of Theorem 1 will be smaller leading to faster convergence compared to an unbiased selection strategy.

**Bias Term.** Similarly, a client selection strategy that prioritizes lower-loss clients will reduce the bias term as $p$ increases. Indeed, $\tilde\rho_b$ should be larger than $\tilde\rho_g$ for a given selection strategy, so decreasing $q$, the number of bad clients selected, decreases the numerator significantly, even as $p$ increases. Likewise, as $p$ increases based on the selection strategy, the denominator increases as well, thereby decreasing the bias term.

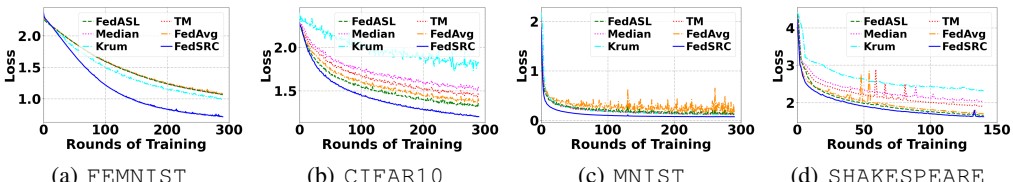

| (a) FEMNIST | (b) CIFAR10 | (c) MNIST | (d) SHAKESPEARE |

Figure 4: Comparison of global loss of FedSRC with benchmark algorithms.

# 4 EVALUATION

## 4.1 SETTINGS

**Dataset and model description.** We utilize four prominent datasets: MNIST LeCun et al. (2010), CIFAR10 Krizhevsky (2009), FEMNIST Caldas et al. (2018), and SHAKESPEARE Caldas et al. (2018) which are widely utilized in the literature McMahan et al. (2017); Li et al. (2020c). For the MNIST and CIFAR10 datasets, we create non-IID settings by assigning each client a dominant class of 50% data and the remaining classes with the rest of the data. FEMNIST and SHAKESPEARE datasets are naturally non-IID. For the handwriting classification of MNIST and FEMNIST, we implement multilayer perceptron (MLP). Convolutional Neural Network(CNN) is used for CIFAR10 image classification, and Recurrent Neural Network (RNN) is used for the next character prediction in SHAKESPEARE. More details of our model description and dataset can be found in Appendix B.

**Evaluation scenarios.** We consider three scenarios reflecting potential data corruption due to sensor quality, malfunction, and aging. *Label shuffling:* It can be referred to as random sensor malfunction, leading to assigning random labels to data. *Label flipping:* It refers to mislabeling data, leading to the same mislabel across all the client data. *Noisy data:* It results from hardware quality in the feature space. To simulate this, we added Gaussian noise to the feature and then clipped the value within the desired feature space level. As the default configuration for our evaluation, we use a mix of 70% good clients with 30% bad clients. The bad clients are equally divided among the three cases. More details of our evaluation scenarios can be found in Appendix B.

**Benchmark algorithms.** To assess the performance of FedSRC, we compare it with the following benchmark algorithms. FedAVG McMahan et al. (2017): the standard federated averaging technique which assigns client weights based on dataset size. Median Yin et al. (2018): a Byzantine robust aggregation rule that independently aggregates each model parameter. For each $i^{th}$ parameter, the server sorts the $i^{th}$ parameters of the selected clients and takes the median as the global parameter. Trimmed Mean Yin et al. (2018): another Byzantine robust aggregation rule that independently aggregates each model parameter. It sorts the parameters and removes a percentage of the largest and smallest values, then averages the remaining for each parameter. FedASL Talukder & Islam (2022): automatically assigns weights to clients based on the median of their training losses. Clients within a predefined "good zone" around the median have higher contributions to the global model, while those outside this zone have inversely proportional contributions of the distance from the median. Krum Blanchard et al. (2017): operates by calculating the Euclidean distance norms between each client's model weights and those of other clients. It removes the highest value for each client, averages the rest, and selects the client with the lowest scores as the next global model.

## 4.2 RESULTS

**Comparison with the benchmark algorithms.** We compare FedSRC with the benchmark algorithms under our default setting. Here, we block 30% clients in FedSRC, Trimmed Mean, and Krum, while FedASL discards clients falling outside one standard deviation (i.e., discards ∼32% clients). FedAVG does not discard any clients. We show the test loss of the global model against an uncorrected test data set in Fig. 4. The comparison of their

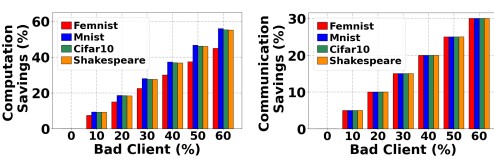

(a) Computation Savings (b) Communication Savings

Figure 5: Client side savings of FedSRC.

accuracy can be found in Fig. 6 in Appendix B. Our experiments reveal that FedSRC consistently outperforms other benchmark algorithms resulting in better global performance. For instance, for the `FEMNIST` data set, after 300 training rounds, FedSRC has 33%, 33%, 33%, 28% and 34% lower loss compared to FedASL, Trimmed Mean, Median, Krum, and FedAVG. Extended evaluation of FedSRC can be found in Appendix B for IID and extreme Non-IID cases for `MNIST` and `CIFAR10` and in all the cases FedSRC outperforms the benchmark algorithms.

**Computation and communication savings.** To evaluate the client-side computation and communication savings achieved by FedSRC, we conduct experiments across various proportions of corrupted clients for different datasets. Clients need to do a forward pass (inference loss) of the first batch only to make the decision of participation. Fig. 5(a) demonstrates that FedSRC yields substantial savings of up to 55% in local computational expenses for higher proportions of malicious clients. Turning to communication savings, when a client abstains from participating, the model upload cost for the trained model is spared. Meanwhile, there is negligible extra communication for sending the test loss to the server. Fig. 5(b) shows that FedSRC can save up to 30% of the communication cost with 60% bad clients. Note here that, unlike computation savings, the communication savings do not depend on the dataset.

**Extended results.** We also evaluate FedSRC's integration with centralized FL approaches and varying degrees of data quality issues. We defer these results to the Appendix C due to space limitations.

## 5 RELATED WORK

The performance of FL deteriorates in the presence of corrupted clients Tolpegin et al. (2020). Notably, FedAVG lacks mechanisms to mitigate the impact of bad clients Fang et al. (2020); Tolpegin et al. (2020). Consequently, various FL algorithms have emerged to defend against data corruption Pillutla et al. (2022); So et al. (2020); Sattler et al. (2020). *Statistics-based algorithms.* Krum Blanchard et al. (2017) selects a global model based on similarity to local models. Bulyan Guerraoui et al. (2018) augments Krum with a Trimmed Mean Yin et al. (2018) variant. However, Bulyan's computational burden arises from dual computations in each training round. *Byzantine Robust Algorithms.* Median Yin et al. (2018) sorts outliers from individual models before global averaging. Geometric Median (GM) Chen et al. (2017); Pillutla et al. (2022) is another technique, but computational intensity hampers its edge feasibility. *Client Selection Algorithms.* Loss-based client selection methods, like AFL Goetz et al. (2019) and Power-of-Choice Cho et al. (2022), assess and prioritize high-loss clients. However, these methods compromise privacy due to ID tagging. *Re-weighting Algorithms.* Zhao et al. (2020) adjusts aggregation weights based on cross-validation. Zhao et al. (2019) and Talukder & Islam (2022) reweight models using auxiliary data, but online detection raises privacy concerns. *Other Data Poisoning Approaches.* Some methods rely on trusted client subsets Cao et al. (2021); Han & Zhang (2020); Li et al. (2020a); Sattler et al. (2020); Ghosh et al. (2020) or cluster-based approaches Sattler et al. (2020) for defense. Yet, trustworthiness and communication constraints pose challenges.

In contrast, FedSRC can manage client-side data corruption without reliance on validation datasets or identity disclosure. Moreover, client-side blocking reduces local computation and communication costs. To the best of our knowledge, FedSRC is the pioneering algorithm that addresses data corruption directly from the client side.

## 6 CONCLUDING REMARKS

In this paper, we presented FedSRC, a novel solution for handling data corruption from the client side to enhance the efficiency of federated learning. Our approach saves communication and computation costs while enhancing global model accuracy and preserving client anonymity. To the best of our knowledge, this is the first attempt to regulate client participation from the client side. **Limitations.** FedSRC relies on client-level statistics to implement its checkpoint, and therefore, it cannot operate if a significant portion of clients is corrupted. FedSRC saves the communication cost of sending a trained model to the server. A client in FedSRC still needs to download the model to check its local test loss, regardless of its participation. As FedSRC works at the client level, it cannot prevent model poisoning attacks with compromised clients. While FedSRC cannot prevent these attacks, it does not introduce any new attack vector.

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

# A    THEORETICAL ANALYSIS

Our proof follows similar lines to that of Cho et al. (2022) but with modifications based on our problem formulation of having good and bad clients as well as our different skewness metrics and local-global objective gap $\rho_g, \rho_b$, and $\Gamma_g$, respectively. To begin, we present some preliminary lemmas that are useful for the proof of Theorem 1.

## A.1    PRELIMINARY LEMMAS

**Lemma 1.** *Assume $F_k$ is L-smooth with global optimum at $w_k^*$. Then for any $w_k$ in the domain of $F_k$,*

$$\|\nabla F_k(w_k)\|^2 \leq 2L(F_k(w_k) - F_k(w_k^*)).$$

*Proof.* Since $F_k$ is $L$-smooth,

$$F_k(w_k) - F_k(w_k^*) - \langle \nabla F_k(w_k^*), w_k - w_k^* \rangle \geq \frac{1}{2L} \|\nabla F_k(w_k) - \nabla F_k(w_k^*)\|^2$$

and $\nabla F_k(w_k^*) = 0$ since $w_k^*$ is a minimizer, so this implies

$$F_k(w_k) - F_k(w_k^*) \geq \frac{1}{2L} \|\nabla F_k(w_k)\|^2$$

which yields the claim. $\square$

**Lemma 2.** *For $w_k^{(t)}$ and $\bar{w}^{(t)} = \frac{1}{m} \sum\limits_{k \in S^{(t)}} w_k^{(t)}$,*

$$\frac{1}{m}\mathbb{E}\left[ \sum_{k \in S^{(t)}} \|\bar{w}^{(t)} - w_k^{(t)}\|^2 \right] \leq 16\eta_t^2 \tau^2 G^2.$$

*Proof.* We have

$$\frac{1}{m}\sum_{k \in S^{(t)}} \|\bar{w}^{(t)} - w_k^{(t)}\|^2 \leq \sum_{k \in S^{(t)}} \|\frac{1}{m}\sum_{\tilde{k} \in S^{(t)}} w_{\tilde{k}}^{(t)} - w_k^{(t)}\|^2 = \frac{1}{m^2}\sum_{k \in S^{(t)}}\sum_{\tilde{k} \in S^{(t)}} \|w_{\tilde{k}}^{(t)} - w_k^{(t)}\|^2$$

$$= \frac{1}{m^2}\sum_{\substack{k,\tilde{k} \in S^{(t)} \\ k \neq \tilde{k}}} \|w_{\tilde{k}}^{(t)} - w_k^{(t)}\|^2,$$

where the inequality follows from $\|\sum_{i=1}^{n} \mathbf{x}_i\|^2 \leq n\sum_{i=1}^{n} \|\mathbf{x}_i\|^2$. For $k = \tilde{k}$, the right hand side of the above inequality is zero. Since the selected clients get updated at every $\tau$ for any $t$ there exist $t_0$ such that $w_{\tilde{k}}^{(t_0)} = w_k^{(t)}$, where $0 \leq t - t_0 \leq \tau$. Hence for any $t$, $\|w_{\tilde{k}}^{(t)} - w_k^{(t)}\|^2$ is bounded above by $\tau$ updates. With non-increasing $\eta_t$ over $t$ and $\eta_{t_0} \leq 2\eta_t$, we can write the right hand side of the above inequality as

$$\frac{1}{m^2}\sum_{\substack{k,\tilde{k} \in S^{(t)} \\ k \neq \tilde{k}}} \|w_{\tilde{k}}^{(t)} - w_k^{(t)}\|^2 \leq \frac{1}{m^2}\sum_{\substack{k,\tilde{k} \in S^{(t)} \\ k \neq \tilde{k}}} \|\sum_{i=t_0}^{t_0+\tau-1} \eta_i \left( g_{\tilde{k}}(w_{\tilde{k}}^{(i)}, \xi_{\tilde{k}}^{(i)}) - g_k^{(i)}(w_k^{(i)}, \xi_k^{(i)}) \right)\|^2$$

$$\leq \frac{\eta_{t_0}^2 \tau}{m^2}\sum_{\substack{k,\tilde{k} \in S^{(t)} \\ k \neq \tilde{k}}}\sum_{i=t_0}^{t_0+\tau-1} \left[ 2\|\left( g_{\tilde{k}}(w_{\tilde{k}}^{(i)}, \xi_{\tilde{k}}^{(i)})\|^2 + 2\|g_k^{(i)}(w_k^{(i)}, \xi_k^{(i)}))\|^2 \right].$$

Taking expectation and applying Assumption 4 gives

$$
\mathbb{E}\Big[\frac{1}{m^2}\sum_{\substack{k,\tilde{k}\in S^{(t)}\\k\neq\tilde{k}}}\|w_{\tilde{k}}^{(t)}-w_k^{(t)}\|^2\Big] \leq \frac{2\eta_{t_0}^2\tau}{m^2}\mathbb{E}\Bigg[\sum_{\substack{k,\tilde{k}\in S^{(t)}\\k\neq\tilde{k}}}\sum_{i=t_0}^{t_0+\tau-1}\Big[\|\big(g_{\tilde{k}}(w_{\tilde{k}}^{(i)},\xi_{\tilde{k}}^{(i)})\|^2+\|g_k^{(i)}(w_k^{(i)},\xi_k^{(i)})\big)\|^2\Big]\Bigg]
$$

$$
\leq \frac{8\eta_t^2\tau}{m^2}\sum_{\substack{k,\tilde{k}\in S^{(t)}\\k\neq\tilde{k}}}\sum_{i=t_0}^{t_0+\tau-1}(G^2+G^2) = \frac{8\eta_t^2\tau}{m^2}\sum_{\substack{k,\tilde{k}\in S^{(t)}\\k\neq\tilde{k}}}2\tau G^2
$$

$$
= \frac{8\eta_t^2\tau}{m^2}m(m-1)2\tau G^2 \leq 16\eta_t^2\tau^2 G^2.
$$

$\square$

**Lemma 3.** *For any random selection strategy, $\mathbb{E}\|\bar{w}^{(t)}-w^*\|^2$ has the following upper bound:*

$$
\mathbb{E}[\|\bar{w}^{(t)}-w^*\|^2] \leq \frac{1}{m}\mathbb{E}[\sum_{k\in S^{(t)}}\|w_k^{(t)}-w^*\|^2].
$$

*Proof.*

$$
\mathbb{E}[\|\bar{w}^{(t)}-w^*\|^2] = \mathbb{E}[\|\frac{1}{m}\sum_{k\in S^{(t)}}w_k^{(t)}-w^*\|^2] = \mathbb{E}[\|\frac{1}{m}\sum_{k\in S^{(t)}}(w_k^{(t)}-w^*)\|^2]
$$

$$
\leq \mathbb{E}[\frac{1}{m}\sum_{k\in S^{(t)}}\|w_k^{(t)}-w^*\|^2].
$$

$\square$

## A.2 PROOF OF THEOREM 1

Letting $\bar{g}(t) = \frac{1}{m}\sum_{k\in S(t)}g_k(w_k^{(t)},\xi_k^{(t)})$, and using the condensed notation $\bar{g}_k = \bar{g}_k(\bar{w}_k^{(t)},\xi_k^{(t)})$ for simplicity, we have

$$
\|\bar{w}^{(t+1)}-w^*\|^2 = \|\bar{w}^{(t)}-\eta_t\bar{g}^{(t)}-w^*\|^2
$$

$$
= \|\bar{w}^{(t)}-\eta_t\bar{g}^{(t)}-w^*-\frac{\eta_t}{m}\sum_{k\in S^{(t)}}\nabla F_k(w_k^{(t)})+\frac{\eta_t}{m}\sum_{k\in S^{(t)}}\nabla F_k(w_k^{(t)})\|^2
$$

$$
= \|\bar{w}^{(t)}-w^*-\frac{\eta_t}{m}\sum_{k\in S^{(t)}}\nabla F_k(w_k^{(t)})\|^2 + \eta_t^2\|\frac{1}{m}\sum_{k\in S^{(t)}}(\nabla F_k(w_k^{(t)})-\bar{g}_k^{(t)})\|^2
$$

$$
+ 2\eta_t\langle\bar{w}^{(t)}-w^*-\frac{\eta_t}{m}\sum_{k\in S^{(t)}}\nabla F_k(w_k^{(t)}),\frac{1}{m}\sum_{k\in S^{(t)}}(\nabla F_k(w_k^{(t)})-\bar{g}_k^{(t)})\rangle
$$

$$
= \|\bar{w}^{(t)}-w^*\|^2 \underbrace{-2\eta_t\langle\bar{w}^{(t)}-w^*,\frac{1}{m}\sum_{k\in S^{(t)}}\nabla F_k(w_k^{(t)})\rangle}_{A_1} +
$$

$$
\underbrace{2\eta_t\langle\bar{w}^{(t)}-w^*-\frac{\eta_t}{m}\sum_{k\in S^{(t)}}\nabla F_k(w_k^{(t)}),\frac{1}{m}\sum_{k\in S^{(t)}}(\nabla F_k(w_k^{(t)})-g_k^{(t)})\rangle}_{A_2} + \underbrace{\eta_t^2\|\frac{1}{m}\sum_{k\in S^{(t)}}\nabla F_k(w_k^{(t)})\|^2}_{A_3}
$$

$$
+ \underbrace{\eta_t^2\|\frac{1}{m}\sum_{k\in S^{(t)}}(\nabla F_k(w_k^{(t)})-\bar{g}_k^{(t)})\|^2}_{A_4}
$$

$$
= \|\bar{w}^{(t)}-w^*\|^2 + A_1 + A_2 + A_3 + A_4. \tag{8}
$$

We first bound the quantity $A_1$ of inequality (8) as follows:

$$
A_1 = -\frac{2\eta_t}{m} \sum_{k \in S^{(t)}} \langle \bar{w}^{(t)} - w^*, \nabla F_k(w_k^{(t)}) \rangle
$$

$$
= -\frac{2\eta_t}{m} \sum_{k \in S^{(t)}} \langle \bar{w}^{(t)} - w_k^{(t)}, \nabla F_k(w_k^{(t)}) \rangle - \frac{2\eta_t}{m} \sum_{k \in S^{(t)}} \langle w_k^{(t)} - w^*, \nabla F_k(w_k^{(t)}) \rangle
$$

$$
\leq \frac{\eta_t}{m} \sum_{k \in S^{(t)}} \left( \frac{1}{\eta_t} \|\bar{w}^{(t)} - w_k^{(t)}\|^2 + \eta_t \|\nabla F_k(w_k^{(t)})\|^2 \right) - \frac{2\eta_t}{m} \sum_{k \in S^{(t)}} \langle w_k^{(t)} - w^*, \nabla F_k(w_k^{(t)}) \rangle
$$

(using the AM-GM and Cauchy–Schwarz inequalities)

$$
= \frac{1}{m} \sum_{k \in S^{(t)}} \|\bar{w}^{(t)} - w_k^{(t)}\|^2 + \frac{\eta_t^2}{m} \sum_{k \in S^{(t)}} \|\nabla F_k(w_k^{(t)})\|^2 - \frac{2\eta_t}{m} \sum_{k \in S^{(t)}} \langle w_k^{(t)} - w^*, \nabla F_k(w_k^{(t)}) \rangle
$$

$$
\leq \frac{1}{m} \sum_{k \in S^{(t)}} \|\bar{w}^{(t)} - w_k^{(t)}\|^2 + \frac{2L\eta_t^2}{m} \sum_{k \in S^{(t)}} \left( F_k(w_k^{(t)}) - F_k^* \right)
$$

$$
- \frac{2\eta_t}{m} \sum_{k \in S^{(t)}} \langle w_k^{(t)} - w^*, \nabla F_k(w_k^{(t)}) \rangle \text{ (using Lemma 1)}
$$

$$
\leq \frac{1}{m} \sum_{k \in S^{(t)}} \|\bar{w}^{(t)} - w_k^{(t)}\|^2 + \frac{2L\eta_t^2}{m} \sum_{k \in S^{(t)}} \left( F_k(w_k^{(t)}) - F_k^* \right)
$$

$$
- \frac{2\eta_t}{m} \sum_{k \in S^{(t)}} \left[ F_k(w_k^{(t)}) - F_k(w^*) \quad + \frac{\mu}{2} \|w_k^{(t)} - w^*\|^2 \right],
$$

where the last inequality follows from $\mu$ strong convexity of $F_k$ (Assumption 2). Hence, by Lemma 2, the expected value of $A_1$ satisfies

$$
\mathbb{E}[A_1] \leq 16\eta_t^2 \tau^2 G^2 - \frac{\eta_t \mu}{m} \mathbb{E}\big[ \sum_{k \in S^{(t)}} \|w_k^{(t)} - w^*\|^2 \big] + \frac{2L\eta_t^2}{m} \mathbb{E}\big[ \sum_{k \in S^{(t)}} \left( F_k(w_k^{(t)}) - F_k^* \right) \big]
$$

$$
- \frac{2\eta_t}{m} \mathbb{E}\big[ \sum_{k \in S^{(t)}} (F_k(w_k^{(t)}) - F_k(w^*)) \big]. \tag{9}
$$

Leaving this bound aside for the moment, next notice that $\mathbb{E}[A_2] = 0$ because of the unbiased gradient assumption (Assumption 3). We may then bound $A_3$ by Lemma 2 as follows:

$$
\mathbb{E}[A_3] = \mathbb{E}\left[ \frac{\eta_t^2}{m^2} \| \sum_{k \in S^{(t)}} \nabla F_k(w_k^{(t)}) \|^2 \right] \leq \frac{\eta_t^2}{m} \sum_{k \in S^{(t)}} \mathbb{E}\left[ \|\nabla F_k(w_k^{(t)})\|^2 \right]
$$

$$
\leq \frac{2L\eta_t^2}{m} \mathbb{E}\left[ \sum_{k \in S^{(t)}} (F_k(w_k^{(t)}) - F_k^*) \right]. \tag{10}
$$

Finally, the bound for $A_4$ is as follows:

$$
\mathbb{E}[A_4] = \mathbb{E}\left[ \frac{\eta_t^2}{m^2} \| \sum_{k \in S^{(t)}} \left( \nabla F_k(w_k^{(t)}) - g_k^{(t)} \right) \|^2 \right] = \frac{\eta_t^2}{m^2} \mathbb{E}_{S^{(t)}}\left[ \sum_{k \in S^{(t)}} \mathbb{E}\|(\nabla F_k(w_k^{(t)}) - g_k^{(t)})\|^2 \right]
$$

$$
\leq \frac{\eta_t^2 m \sigma^2}{m^2} = \frac{\eta_t^2 \sigma^2}{m}, \tag{11}
$$

where the second equality and inequality use Assumption 3.

Using the bounds (9), (10), and (11) in (8), we have

$$
\mathbb{E}[\|\bar{w}^{(t+1)} - w^*\|^2] \leq \mathbb{E}[\|\bar{w}^{(t)} - w^*\|^2] + \sum_{i=1}^{4} \mathbb{E}[A_i] \leq \mathbb{E}\big[\|\bar{w}^{(t)} - w^*\|^2\big] - \frac{\eta_t \mu}{m} \mathbb{E}\big[ \sum_{k \in S^{(t)}} \|w_k^{(t)} - w^*\|^2 \big]
$$

$$+ 16\eta_t^2 \tau^2 G^2 + \frac{\eta_t^2 \sigma^2}{m} + \frac{4L\eta_t}{m} \mathbb{E}\big[ \sum_{k \in S^{(t)}} (F_k(w_k^{(t)}) - F_k^*)\big] - \frac{2\eta_t}{m} \mathbb{E}\big[ \sum_{k \in S^{(t)}} (F_k(w_k^{(t)}) - F_k(w^*))\big]$$

$$\leq (1 - \eta_t \mu) \mathbb{E}[\|\bar{w}^{(t)} - w^*\|^2] + 16\eta_t^2 \tau^2 G^2 + \frac{\eta_t^2 \sigma^2}{m} + \underbrace{\frac{4L\eta_t^2}{m} \mathbb{E}\big[ \sum_{k \in S^{(t)}} (F_k(w_k^{(t)}) - F_k^*)\big]}_{A_5}$$

$$\underbrace{- \frac{2\eta_t}{m} \mathbb{E}\big[ \sum_{k \in S^{(t)}} (F_k(w_k^{(t)}) - F_k(w^*))\big]}_{A_5}. \tag{12}$$

The final inequality above utilizes Lemma 3.

Now we bound $A_5$ as follows:

$$A_5 = \mathbb{E}\big[ \frac{4L\eta_t^2}{m} \sum_{k \in S^{(t)}} F_k(w_k^{(t)}) - \frac{2\eta_t}{m} \sum_{k \in S(t)} F_k(w_k^{(t)}) - \frac{2\eta_t}{m} \sum_{k \in S^{(t)}} (F_k^* - F_k(w^*))$$

$$+ \frac{2\eta_t}{m} \sum_{k \in S^{(t)}} F_k^* - \frac{4L\eta_t^2}{m} \sum_{k \in S^{(t)}} F_k^* \big]$$

$$= \mathbb{E}\big[ \underbrace{\frac{2\eta_t(2L\eta_t - 1)}{m} \sum_{k \in S^{(t)}} (F_k(w_k^{(t)}) - F_k^*)}_{A_6}\big] + 2\eta_t E\big[\frac{1}{m} \sum_{k \in S(t)} (F_k(w^*) - F_k^*)\big].$$

Take $\eta_t < 1/(4L)$ and define $\upsilon_t = 2\eta_t(1 - 2L\eta_t) \geq 0$; then we can bound $A_6$ as

$$- \frac{\upsilon_t}{m} \sum_{k \in S^{(t)}} (F_k(w_k^{(t)}) - F_k^*)$$

$$= - \frac{\upsilon_t}{m} \sum_{k \in S^{(t)}} (F_k(w_k^{(t)}) - F_k(\bar{w}^{(t)}) + F_k(\bar{w}^{(t)}) - F_k^*)$$

$$= - \frac{\upsilon_t}{m} \sum_{k \in S^{(t)}} \big[F_k(w_k^{(t)}) - F_k(\bar{w}^{(t)})\big] - \frac{\upsilon_t}{m} \sum_{k \in S^{(t)}} \big[F_k(\bar{w}^{(t)}) - F_k^*\big]$$

$$\leq - \frac{\upsilon_t}{m} \sum_{k \in S^{(t)}} \big[\langle \nabla F_k(w^{(t)}), w_k^{(t)} - \bar{w}^{(t)} \rangle + \frac{\mu}{2}\|w_k^{(t)} - \bar{w}^{(t)}\|^2\big] - \frac{\upsilon_t}{m} \sum_{k \in S^{(t)}} \big[F_k(\bar{w}^{(t)}) - F_k^*\big]$$

$$\leq \frac{\nu_t}{m} \sum_{k \in S^{(t)}} \big[\eta_t L(F_k(\bar{w}^{(t)}) - F_k^*) + (\frac{1}{2\eta_t} - \frac{\mu}{2})\|w_k^{(t)} - \bar{w}^{(t)}\|^2\big] - \frac{\upsilon_t}{m} \sum_{k \in S^{(t)}} \big[F_k(\bar{w}^{(t)}) - F_k^*\big]$$

(using the Cauchy-Schwarz inequality, the AM-GM inequality, and Lemma 1)

$$= - \frac{\nu_t}{m}(1 - \eta_t L) \sum_{k \in S^{(t)}} (F_k(\bar{w}^{(t)}) - F_k^*) + \Big(\frac{\nu_t}{2\eta_t m} - \frac{\nu_t \mu}{2m}\Big) \sum_{k \in S^{(t)}} \|w_k^{(t)} - \bar{w}^{(t)}\|^2$$

$$\leq - \frac{\nu_t}{m}(1 - \eta_t L) \sum_{k \in S^{(t)}} (F_k(\bar{w}^{(t)}) - F_k^*) + \frac{1}{m} \sum_{k \in S^{(t)}} \|w_k^{(t)} - \bar{w}^{(t)}\|^2. \tag{13}$$

The first inequality above uses $\mu$ strong convexity of $F_k$, the subsequent inequality uses $L$–smoothness of $F_k$, and the final inequality follows because $\frac{\nu_t(1 - \eta_t \mu)}{2\eta_t} \leq 1$. Hence, we can bound $A_5$ as follows:

$$\mathbb{E}[A_5] \leq -\frac{\nu_t}{m}(1 - \eta_t L)\mathbb{E}\Big[\sum_{k \in S^{(t)}}(F_k(\bar{w}^{(t)}) - F_k^*)\Big] + \frac{1}{m}\mathbb{E}\Big[\sum_{k \in S^{(t)}}\|w_k^{(t)} - \bar{w}^{(t)}\|^2\Big]$$

$$+ \frac{2\eta_t}{m}\mathbb{E}\Big[\sum_{k \in S^{(t)}}(F_k(w^*) - F_k^*)\Big]$$

$$\leq -\frac{\nu_t}{m}(1 - \eta_t L)\mathbb{E}\Big[\sum_{k \in S^{(t)}}(F_k(\bar{w}^{(t)}) - F_k^*)\Big] + 16\eta_t^2 \tau^2 G^2 + \frac{2\eta_t}{m}\mathbb{E}\Big[\sum_{k \in S^{(t)}}(F_k(w^*) - F_k^*)\Big]$$

$$= -\frac{\nu_t}{m}(1 - \eta_t L)\mathbb{E}\Big[\sum_{k \in S^{(t)} \cap \mathcal{G}}(F_k(\bar{w}^{(t)}) - F_k^*) + \sum_{k \in S^{(t)} \cap \mathcal{B}}(F_k(\bar{w}^{(t)}) - F_k^*)\Big] + 16\eta_t^2 \tau^2 G^2$$

$$+ \frac{2\eta_t}{m}\mathbb{E}\Big[\sum_{k \in S^{(t)} \cap \mathcal{G}}(F_k(w^*) - F_k^*) + \sum_{k \in S^{(t)} \cap \mathcal{B}}(F_k(w^*) - F_k^*)\Big]$$

$$= 16\eta_t^2 \tau^2 G^2 - \frac{\nu_t(1 - \eta_t L)}{m}\mathbb{E}\Big[\big(p\rho_g(S(\pi, \bar{w}^{(\tau\lfloor t/\tau \rfloor)}), \bar{w}^{(t)})$$

$$+ q\rho_b(S(\pi, \bar{w}^{(\tau\lfloor t/\tau \rfloor)}), \bar{w}(t))\big)(F_g(\bar{w}^{(t)}) - \sum_{k \in \mathcal{G}}p_k F_k^*\Big] + \frac{2\eta_t}{m}\mathbb{E}\Big[\big(p\rho_g(S(\pi, \bar{w}^{(\tau\lfloor t/\tau \rfloor)}), w^*)$$

$$+ q\rho_b(S(\pi, \bar{w}^{(\tau\lfloor t/\tau \rfloor)}), w^*)(F_g(w^*) - \sum_{k \in \mathcal{G}}p_k F_k^*\Big]$$

$$\leq 16\eta_t^2 \tau^2 G^2 \underbrace{- \frac{\nu_t(1 - \eta_t L)}{m}\big[p\bar{\rho}_g + q\bar{\rho}_b\big]\big(\mathbb{E}[F_g(\bar{w}^{(t)} - \sum_{k \in \mathcal{G}}p_k F_k^*\big)\big)}_{A_7} + \frac{2\eta_t}{m}(p\tilde{\rho}_g + q\tilde{\rho}_b)\Gamma_g$$

$$\tag{14}$$

We used the definition of $\rho(S(\pi, w), w')$ and $\Gamma_g$ to arrive at (14). We can get a bound for $A_7$ in (14) as follows:

$$A_7 = -\frac{\nu_t(1 - \eta_t L)}{m}\big[p\bar{\rho}_g + q\bar{\rho}_b\big]\sum_{k \in \mathcal{G}}p_k\big(\mathbb{E}[F_k(\bar{w}^{(t)})] - F^* + F^* - F_k^*\big)$$

$$= -\frac{\nu_t(1 - \eta_t L)}{m}\big[p\bar{\rho}_g + q\bar{\rho}_b\big]\sum_{k \in \mathcal{G}}p_k\left(\mathbb{E}[F_k(\bar{w}^{(t)})] - F^* + F^* - F_k^*\right)$$

$$= -\frac{\nu_t(1 - \eta_t L)}{m}\big[p\bar{\rho}_g + q\bar{\rho}_b\big]\sum_{k \in \mathcal{G}}p_k\big(\mathbb{E}[F_k(\bar{w}^{(t)})] - F^*\big)$$

$$- \frac{\nu_t(1 - \eta_t L)}{m}\big[p\bar{\rho}_g + q\bar{\rho}_b\big]\sum_{k \in \mathcal{G}}p_k\big(F^* - F_k^*\big)$$

$$= -\frac{\nu_t(1 - \eta_t L)}{m}\big[p\bar{\rho}_g + q\bar{\rho}_b\big]\big(\mathbb{E}[F_g(\bar{w}^{(t)})] - F^*\big) - \frac{\nu_t(1 - \eta_t L)}{m}\big[p\bar{\rho}_g + q\bar{\rho}_b\big]\Gamma_g$$

(using the definition of $\Gamma_g$)

$$\leq -\frac{\nu_t(1 - \eta_t L)\mu\big[p\bar{\rho}_g + q\bar{\rho}_b\big]}{2m}\mathbb{E}\big[\|\bar{w}^{(t)} - w^*\|^2\big] - \frac{\nu_t(1 - \eta_t L)}{m}\big[p\bar{\rho}_g + q\bar{\rho}_b\big]\Gamma_g$$

(using $\mu$ strongly convexity)

$$= -\frac{2\eta_t(1 - 2L\eta_t)(1 - \eta_t L)\mu\big[p\bar{\rho}_g + q\bar{\rho}_b\big]}{2m}\mathbb{E}\big[\|\bar{w}^{(t)} - w^*\|^2\big]$$

$$- \frac{2\eta_t(1 - 2L\eta_t)(1 - \eta_t L)}{m}\big[p\bar{\rho}_g + q\bar{\rho}_b\big]\Gamma_g$$

$$\leq -\frac{3\eta_t\mu\big[p\bar{\rho}_g + q\bar{\rho}_b\big]}{8m}\mathbb{E}\big[\|\bar{w}^{(t)} - w^*\|^2\big] - \frac{2\eta_t\big[p\bar{\rho}_g + q\bar{\rho}_b\big]\Gamma_g}{m} + \frac{6\eta_t^2\big[p\bar{\rho}_g + q\bar{\rho}_b\big]L\Gamma_g}{m} \tag{15}$$

where equation (15) is due to $\mu$ strong convexity and we used $-2\eta_t(1 - 2L\eta_t)(1 - L\eta_t) \leq -\frac{3}{4}\eta_t$ and $-(1 - 2L\eta_t)(1 - L\eta_t) \leq -(1 - 3L\eta_t)$. Hence, the bound of $A_5$ is as follows:

$$
\frac{4L\eta_t}{m}\mathbb{E}\Big[\sum_{k \in S(t)}\big[(F_k(w_k^{(t)}) - F_k^*) - \frac{2\eta_t}{m}(F_k(w_k^{(t)} - F_k(w^*))]\big]
$$

$$
\leq -\frac{3\eta_t\mu[p\bar{\rho}_g + q\bar{\rho}_b]}{8m}\mathbb{E}\big[\|\bar{w}^{(t)} - w^*\|^2 + \eta_t^2\left(\frac{6[p\bar{\rho}_g + q\bar{\rho}_b]L\Gamma_g}{m} + 16\tau^2 G^2\right)
$$

$$
- \frac{2\eta_t[p\bar{\rho}_g + q\bar{\rho}_b]\Gamma_g}{m} + \frac{2\eta_t[p\tilde{\rho}_g + q\tilde{\rho}_b]\Gamma_g}{m}. \tag{16}
$$

Finally, using equation (12), and (16) we can bound $\|\bar{w}^{(t+1)} - w^*\|$ as follows:

$$
\mathbb{E}\big[\|\bar{w}^{(t+1)} - w^*\|\big] \leq \big[1 - \eta_t\mu\big[1 + \frac{3(p\bar{\rho}_g + q\bar{\rho}_b)}{8m}\big]\big]\mathbb{E}\big[\bar{w}^{(t)} - w^*\|^2\big]
$$

$$
+ \eta_t^2\big[32\tau^2 G^2 + \frac{\sigma^2}{m} + \frac{6(p\bar{\rho}_g + q\bar{\rho}_b)L\Gamma_g}{m}\big] + \frac{2\eta_t\Gamma_g}{m}(p\tilde{\rho}_g + q\tilde{\rho}_b - p\bar{\rho}_g - q\bar{\rho}_b)
$$

$$
\leq \big[1 - \eta_t\mu\big[1 + \frac{3(p\bar{\rho}_g + q\bar{\rho}_g)}{8m}\big]\big]\mathbb{E}\big[\bar{w}^{(t)} - w^*\|^2\big] + \eta_t^2\big[32\tau^2 G^2 + \frac{\sigma^2}{m} + \frac{6(p\bar{\rho}_b + q\bar{\rho}_b)L\Gamma_g}{m}\big]
$$

$$
+ \frac{2\eta_t\Gamma}{m}(p\tilde{\rho}_g + q\tilde{\rho}_b - p\bar{\rho}_g - q\bar{\rho}_g). \tag{17}
$$

Equation (17) is obtained using $\bar{\rho}_g \leq \bar{\rho}_b$, which gives

$$
\mathbb{E}\big[\|\bar{w}^{(t+1)} - w^*\|\big] \leq \big[1 - \eta_t\mu\big[1 + \frac{3\bar{\rho}_g}{8}\big]\big]\mathbb{E}\big[\bar{w}^{(t)} - w^*\|^2\big]
$$

$$
+ \eta_t^2\big[32\tau^2 G^2 + \frac{\sigma^2}{m} + 6\bar{\rho}_b L\Gamma_g\big] + \frac{2\eta_t\Gamma_g}{m}(p\tilde{\rho}_g + q\tilde{\rho}_b - m\bar{\rho}_g).
$$

By setting $\Delta_{t+1} = \mathbb{E}\big[\|\bar{w}^{(t+1)} - w^*\|^2\big]$, $B = 1 + \frac{3\bar{\rho}_g}{8}$, $C = 32\tau^2 G^2 + \frac{\sigma^2}{m} + 6\bar{\rho}_b L\Gamma_g$, $D = \frac{2\Gamma_g}{m}(p\tilde{\rho}_g + q\tilde{\rho}_b - m\bar{\rho}_g)$, we get

$$
\Delta_{t+1} \leq (1 - \eta_t\mu B)\Delta_t + \eta_t^2 C + D\eta_t.
$$

For a decreasing stepsize, $\eta_t = \frac{\beta}{t+\gamma}$ for some $\beta > \frac{1}{\mu B}, \gamma > 0$, we have that $\Delta_t \leq \frac{\psi}{t+\gamma}$, where

$$
\psi = \max\left\{(\gamma + 1)\|\bar{w}^{(1)} - w^*\|^2, \frac{\beta^2 C + \beta D(t + \gamma)}{\beta\mu B - 1}\right\}.
$$

This can be shown by induction on $t$ (see Lemma 4 below). Then using the $L$−smoothness of $F(\cdot)$ we get

$$
\mathbb{E}\big[F(\bar{w}^{(t)}] - F^* \leq \frac{L}{2}\Delta_t \leq \frac{L}{2}\frac{\psi}{\gamma + t}.
$$

Now for $\beta = \frac{1}{\mu}$, we get

$$
\mathbb{E}[F(\bar{w}^{(T)})] - F^* \leq \frac{1}{(T + \gamma)}\left[\frac{4L(32\tau^2 G^2 + \sigma^2/m)}{3\mu^2\bar{\rho}_g} + \frac{8L^2\Gamma_g}{\mu^2}\frac{\bar{\rho}_b}{\bar{\rho}_g} + \frac{L(\gamma + 1)(\|\bar{w}^{(1)} - w^*\|^2)}{2}\right]
$$

$$
+ \frac{8L\Gamma_g}{3\mu}\left(\frac{p\tilde{\rho}_g + q\tilde{\rho}_b}{m\bar{\rho}_g} - 1\right),
$$

which completes the proof of the theorem. $\qquad\square$

**Lemma 4.** *For a decreasing stepsize, $\eta_t = \frac{\beta}{t+\gamma}$ for some $\beta > \frac{1}{\mu B}$, $\gamma > 0$,*

$$\Delta_t \leq \frac{\psi}{t+\gamma} \tag{18}$$

*where,*

$$\psi = \max\Big\{(\gamma+1)\|\bar{w}^{(1)} - w^*\|^2, \frac{1}{\beta\mu B - 1}(\beta^2 C + D\beta(t+\gamma))\Big\} \tag{19}$$

*and*

$$\Delta_{t+1} \leq (1 - \eta_t \mu B)\Delta_t + \eta_t^2 C + \eta_t D.$$

*Proof.* For $t = 1$, equation (18) holds clearly as (using (19))

$$\Delta_1 \leq \frac{\psi}{\gamma+1} \leq \|\bar{w}^{(1)} - w^*\|^2 = \Delta_1$$

Assume that it holds for some $t$, then

$$\begin{aligned}
\Delta_{t+1} &\leq (1 - \eta_t \mu B)\Delta_t + \eta_t^2 C + \eta_t D \\
&\leq (1 - \frac{\beta}{t+\gamma}\mu B)\frac{\psi}{t+\gamma} + \frac{\beta^2}{(t+\gamma)^2}C + \frac{\beta}{t+\gamma}D \\
&= \frac{t+\gamma - \beta\mu B}{(t+\gamma)^2}\psi + \frac{\beta^2 C + \beta D(t+\gamma)}{(t+\gamma)^2} \\
&= \frac{t+\gamma-1}{(t+\gamma)^2}\psi + \frac{\beta^2 C + \beta D(t+\gamma)}{(t+\gamma)^2} - \frac{\beta\mu B - 1}{(t+\gamma)^2}\psi \\
&= \frac{t+\gamma-1}{(t+\gamma)^2}\psi \qquad \text{(Using (19))} \\
&\leq \frac{t+\gamma-1}{(t+\gamma)^2 - 1}\psi = \frac{\psi}{t+\gamma+1}
\end{aligned}$$

$\square$

## B  DATASET AND MODEL DESCRIPTION - EXTENDED

### B.1  DATASETS

We utilize four prominent datasets: `MNIST`, `CIFAR10`, `FEMNIST`, and the `SHAKESPEARE` dataset, widely referenced in the literature McMahan et al. (2017); Li et al. (2020c).

**`MNIST` LeCun et al. (2010).** Renowned for handwriting recognition, this dataset consists of 70,000 gray-scale $28 \times 28$ images. It includes 60,000 training samples and 10,000 test samples, spanning ten classes (digits 0-9). We distribute `MNIST` training data evenly among 100 clients for the IID case. For Non-IID, each client possesses one dominant class with 80% of the data, while the remaining classes share 20%. In the extreme Non-IID scenario, a class contributes data to at most two clients. The standard test set evaluates global model performance.

**`CIFAR10` Krizhevsky (2009).** Comprising 60,000 color $32 \times 32$ images, the `CIFAR10` dataset encompasses 50,000 training images and 10,000 test images across ten classes. Similar to `MNIST`, we consider three distribution types: IID, Non-IID, and extreme Non-IID. Dividing the dataset into 100 clients, each IID client receives 500 samples. For Non-IID scenarios, one dominant class constitutes 80% of a client's data, while the rest is shared among other classes. In the extreme Non-IID case, each class contributes data to a maximum of two clients. The test set is used to evaluate the performance of the global model

**`FEMNIST` Caldas et al. (2018).** Derived from the LEAF dataset and implemented using Tensor-Flow Federated, `FEMNIST` involves 3,383 unique users (first 1000 used). It offers 341,873 training

examples and 40,832 test examples, featuring gray-scale $28 \times 28$ images. The dataset creates a non-IID and heterogeneous setting, with each user representing a distinct client. Test sets from distinct clients are collected together to evaluate global performance.

**SHAKESPEARE Caldas et al. (2018).** Based on *"The Complete Works of William Shakespeare"*, this dataset uses speaking roles in plays to represent individual clients. It encompasses 715 genuine users (71 clients with at least 60 test data points), providing 16,068 training examples and 2,356 test examples in text format. Like FEMNIST, the SHAKESPEARE dataset is non-IID and heterogeneous, associating each user with a unique client.

Table 1: Dataset and Model

| Dataset | Training | Test | #Client | Distribution | Model |
|---|---|---|---|---|---|
| MNIST | 60,000 | 10,000 | 300 | IID/Non-IID | LR |
| CIFAR10 | 50,000 | 10,000 | 300 | IID/Non-IID | CNN |
| FEMNIST | 341,873 | 40,832 | 3383 | Non-IID | LR |
| SHAKESPEARE | 16,068 | 2,356 | 715 | Non-IID | RNN |

## B.2    MODEL PARAMETERS

In the context of an edge setup with IoT devices as clients, we prioritize lightweight models to accommodate limited power and computational capabilities.

**MNIST.** For the MNIST dataset, we adopt a simple Multi-Layer Perceptron (MLP) classifier using TensorFlow Keras. The architecture includes two hidden layers with ReLU activation: one with 200 neurons, the other with 100 neurons. An output layer with 10 neurons and softmax activation handles classification. Input features are flattened, and labels are one-hot encoded. The model employs the Adam optimizer with a learning rate of 0.001 and categorical cross-entropy loss. Training spans 300 epochs for all distribution cases.

**CIFAR10.** Employing the CIFAR10 dataset, we employ a lightweight Convolutional Neural Network (CNN) classifier using TensorFlow Keras. The CNN architecture involves two sets of convolutional layers, max-pooling layers, dropout layers, and fully connected layers. ReLU activation functions operate in the convolutional layers, while softmax is used for the output layer. The model employs categorical cross-entropy loss, the Adam optimizer with varying learning rates, and trains until 300 rounds.

**FEMNIST.** Addressing the FEMNIST dataset, we use a simple MLP with two hidden layers. These layers consist of fully connected dense layers with ReLU activation. The model input shape is 784 (pixels in each image), featuring 64 neurons in the first hidden layer. The output layer, with 10 neurons, lacks an activation function to complement the Sparse-Categorical-Crossentropy loss function. The optimization employs a learning rate of 0.001. Training spans 300 epochs.

**SHAKESPEARE.** Utilizing the SHAKESPEARE dataset, we deploy a Recurrent Neural Network (RNN) featuring a GRU layer with stateful=True. Input data is preprocessed via an ASCII character-to-index lookup table, forming sequences of length 50+1. The architecture integrates an embedding layer, a GRU layer with varying units, and a dense layer with 86 output units. A custom evaluation metric gauges the model's character prediction accuracy across the input sequence. There are 150 rounds of training across the clients.

**Evaluation Scenarios** We consider three different scenarios that reflect potential data corruption due to sensor quality and aging:

**Label shuffling.** In this scenario, we consider sensors' label interpretations are incorrect, leading to the assignment of random labels to data. We experiment with varying percentages of clients whose labels are randomly shuffled.

**Label flipping.** Here, a random label is assigned to each client, with the same labels across its data (e.g., all of Client 1's data is labeled 2). We consider a fraction of sensors that consistently produce a fixed, random label output.

**Noisy data.** This scenario involves correct label interpretation but noisy feature spaces. To simulate this, we introduce Gaussian noise to the features. For selected clients, the input data is first normalized to $[0, 1]$ and then we add Gaussian noise $x = x + \epsilon$, where $\epsilon \sim N(0, 0.7)$. The resulting values are clipped again to $[0, 1]$.

## C EVALUATION - EXTENDED

### C.1 COMPARISON WITH BENCHMARK ALGORITHMS - EXTENDED

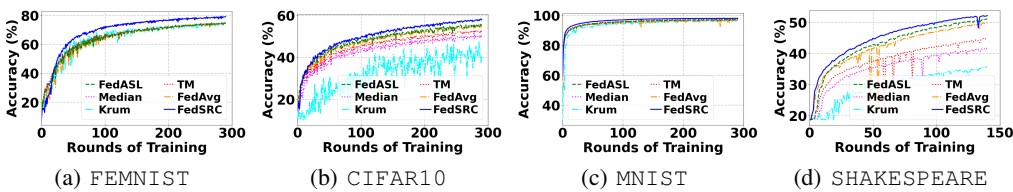

Figure 6: Comparison of global accuracy of FedSRC with other state of the arts algorithm for the FEMNIST, CIFAR10, MNIST and SHAKESPEARE datasets.

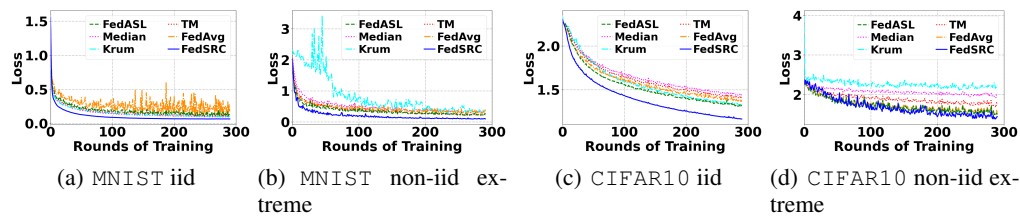

Figure 7: Comparison of loss of FedSRC with other state-of-the-arts algorithm for the CIFAR10 and MNIST datasets.

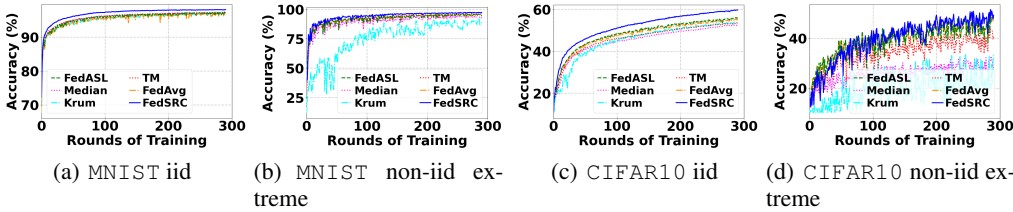

Figure 8: Comparison of accuracy of FedSRC with other state-of-the-arts algorithm for the CIFAR10 and MNIST datasets.

To comprehensively evaluate the effectiveness of FedSRC compared to state-of-the-art algorithms, we have conducted extensive assessments across diverse datasets, including FEMNIST, CIFAR10, MNIST, and SHAKESPEARE. These evaluations were carried out under our default settings, involving 30% data corruption. In this context, we present an in-depth evaluation focusing on MNIST and CIFAR10 datasets, considering both the IID (Independent and Identically Distributed) and Non-IID extreme cases.

In our evaluation, we blocked 30% of clients in FedSRC, Trimmed Mean, and Krum algorithms. In contrast, FedASL excludes clients falling outside one standard deviation, which accounts for discarding approximately 32% of clients. Notably, FedAVG does not discard any clients. The performance metrics displayed are the loss and accuracy of the global model when assessed against the test dataset.

Specifically, we present the outcomes in Figs. 7 represent the loss plot and 8 and 6 represent the accuracy plot. Our experiments reveal that FedSRC consistently outperforms other benchmark algorithms resulting in better global performance in the presence of corrupted clients.

### C.2 INTEGRATION WITH EXISTING ALGORITHMS

We demonstrate the effectiveness of integrating FedSRC with other algorithms by implementing it at the client level while maintaining aggregation protocols such as FedAVG, FedASL, Trimmed Mean,

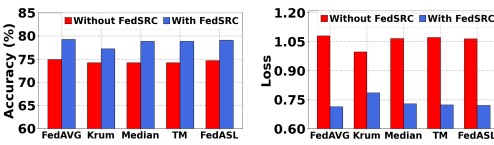

Figure 9: Performance comparison with integrating FedSRC with existing algorithms for `FEMNIST`.

Krum, and Median on the server side. As shown in Fig. 9, our integration approach enhances the performance of these pre-existing algorithms (about 6% increase in accuracy) and reduces the error loss (about 33% decrease in loss) in the presence of unreliable clients all the while also reducing computation and communication costs.

## C.3 SENSITIVITY ANALYSIS

**Impact of blocking percentage.** To understand the effects of user-defined blocking percentage, we evaluate the `FEMNIST` dataset with 30% data corruption. We vary client blocking from 0% to 90%. The goal is to find how the performance FedSRC is impacted by different degrees of client exclusion. The results, as depicted in Fig. 10, show that the optimal performance achieved when correctly estimating a 30% threshold. However, there is no significant degradation, especially when overestimating the blocking percentage.

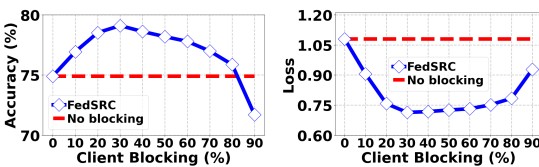

Figure 10: Effect of our cutoff (range) in performance of FedSRC for `FEMNIST` dataset.

The results highlight the robustness of our approach.

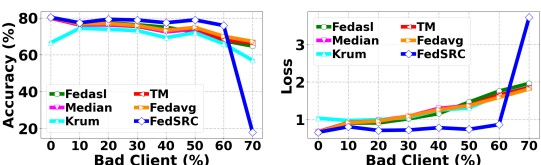

Figure 11: Performance comparison of FedSRC with other algorithms in the presence of different percentages of bad clients for `FEMNIST` dataset in shuffling.

**Impact of Different Percentage of Bad Client:** To assess our algorithm against varying levels of corrupted data, we use `FEMNIST` dataset with different percentages of bad clients and set the client blocking parameters of FedSRC and benchmark algorithms. Fig. 11 shows that as the percentage of unreliable clients increases, conventional algorithms' accuracy declines. In contrast, our FedSRC demonstrates remarkable robustness, effectively managing up to 60% of clients with erroneous behavior. Naturally, as our algorithm utilizes clients' loss statistics, its performance falters drastically with a higher percentage of bad clients.

