# OpenReview forum: "FedSRC: Federated Learning with Self-Regulating Clients"
_ICLR.cc/2024/Conference — Submitted to ICLR 2024_

### Official Review · Reviewer_Xw96 · 2023-10-20

**Soundness:** 2 fair
**Presentation:** 3 good
**Contribution:** 2 fair
**Rating:** 3
**Confidence:** 4

**Summary:**

This paper proposes a client-side selection method named FedSRC to alleviate the impact of clients with “bad” data and improve the overall performance of federated learning. The proposed method utilizes a lightweight estimator on each client to determine whether the client needs to train and upload the model in each round. The empirical results show that FedSRC attains faster convergence and lower loss.

**Strengths:**

This paper proposes a novel method to tackle the statistical heterogeneity in federated learning by determining the participation of each client on the client side. Particularly,

1.	The client-side selection method is lightweight and easy to deploy.

2.	Theoretical analysis for the convergence of FedSRC is provided, and empirical results also support the effectiveness of FedSRC with the presence of “bad” data.

**Weaknesses:**

Though FedSRC shows good results in the special scenarios given in this paper, the motivations announced by the authors do not make sense to me. Specifically,

1.	The authors claim that FedSRC can maintain the anonymity of clients compared with adaptive client selection methods. However, as the clients need to communicate with the server with network protocols, the server can always collect the IP address of each client, which makes it not practical to maintain anonymity in federated learning even with FedSRC.

2.	The authors claim that only the clients with low inference losses participate in the training, which is irrational and may violate fairness significantly. The clients with low losses are trained and then attain lower losses in the next round, while the clients with high losses are never trained and attain higher losses in the next round. As a result, clients with high losses will never participate in training, which makes the global model highly biased. Accordingly, FedSRC may not have good performance on general scenarios beyond the settings used in the paper.

3.	Client-side selection is more vulnerable to malicious clients since the server cannot control the participation at all. This makes the Byzantine Attack easier. Although the authors claim that their method can be combined with server-side defenses, the server may need to spend more effort in defending against the adaptive attacks targeting FedSRC. The paper does not provide any results about the robustness of FedSRC combined with other defense methods.

4.	The authors claim that FedSRC can save the communication overhead. However, FedSRC requires each client to download the global model in every round, which is not required by standard federated learning algorithms. This part will significantly increase the communication round compared to FedAVG with partial participation.

**Questions:**

1.	How does FedSRC distinguish clients with “bad” data and clients with rare data that is different from the data of other clients?

2.	What is the definition of anonymity? Is the IP address considered as an identity?

---

### Official Review · Reviewer_fQ8U · 2023-10-30

**Soundness:** 3 good
**Presentation:** 2 fair
**Contribution:** 1 poor
**Rating:** 3
**Confidence:** 4

**Summary:**

The paper tackles the problem of performance degradation due to statistical heterogeneity and varying data quality across clients,  in he context of federated learning. The paper focuses on a byzantine federated learning scenario, where a fraction of the clients (referred to as ``bad clients'') have corrupted or low quality data.

Differently from  prior lines of work that unnecessary uses client's resources or compromise their anonymity---typically through assigning lower weights to the bad clients during aggregation, or by means of ``active client selection''---this paper introduces a novel self-regulating client participation policy called FedSRC.

FedSRC employs a simple client selection strategy: clients with inference loss (computed using the global model) lower  than a given threshold (computed globally at every round) participate in the training of the global model.

The paper proves theoretically the convergence of federated learning with self-regulating clients to the optimal solution under standard smoothness, strong-convexity, stochastic gradients unbiasedness and bounded variance.

Finally, the paper conducts numerical simulations to compare FedSRC to other federated learning paradigms, and to quantify the computation and communication savings achieved by FedSRC.

**Strengths:**

1. The paper effectively motivates the identified problem and  highlights the shortcomings of previous research in this domain.

2. The paper introduces a straightforward and resource-efficient method to address the adverse effects caused by "bad clients" in the context of federated learning. The simplicity and lightweight nature of the proposed approach contribute to its appeal.

3. The paper demonstrates a commendable level of transparency by acknowledging the limitations of the FedSRC approach. It conscientiously discusses scenarios where this method might encounter challenges or prove less effective, enhancing the overall credibility of the research.

**Weaknesses:**

* The paper has a list of questionable claims:
    * **Page 2.** "FedSRC offers the first variation of FL that allows clients to make strategic decisions to aid the FL global model." Many other works (Tu et al., 2022; Donahue et al., 2021; Cho et al., 202) study the problem of federated learning when clients can opt-out of federation.
    * **Page 3.** "treating every client equally (e.g., model aggregation of FedAVG)." The original vanilla-FedAvg algorithm considers weighted aggregation.
* The conclusions obtained from Figure 2 and Figure 3 might be misleading. They are obtained in the particular scenarios where bad clients have noisy observations. For example, the same conclusions may not hold in the label flipping scenario.
* The modelling of  the problem, and some of the definitions  are not always clear.
    * Definition 1 is informal. It does not precise what does inclusion means, and  what the "converged global objective" refers to (is it the global objective $F$ (defined in (1)) or the objective of the good clients $F_g$?).
    * I am unsure how (2) is obtained. It seems that the paper assumes that $w^* = w_{g}^{*}$. Overall, I have the impression that the paper confuses $F$ with $F_g$.
* The technical novelty of the paper is not clear. The theoretical results of this paper are an8 application of the convergence analysis of federated   optimization for biased client selection strategies, first presented in (Cho et al., 2022).
* The description of the experimental settings does not precise if the loss/accuracy are computed on the good clients only, or on all the clients. Moreover, it does not precise how  the individual clients' losses are averaged.
* Other minor concerns:
   * The paper cites other references using the author mode, when it should use the parenthesis mode. For example, the sentence *...like Google's Board Hard et al. (2019)...* should be written *...like Google's Board (Hard et al., 2019)...*
   * Sometimes, the paper makes a mistake in the usage of the \ref command, e.g., Eq. equation 1 (in Section 2.1).
   * When the paper introduces the shadow sequence $\bar{w}$ at the end of Section 2.1, it should mention that this technique was introduced in prior works, and cite them.


----
References
- Xuezhen Tu, Kun Zhu, Nguyen Cong Luong, Dusit Niyato, Yang Zhang, and Juan Li. "Incentive Mechanisms for Federated Learning: From Economic and Game Theoretic Perspective". In: IEEE Transactions on Cognitive Communications and Networking 8.3 (2022), pp. 1566–1593.
- Kate Donahue and Jon Kleinberg. "Model-sharing Games: Analyzing Federated Learning Under Voluntary Participation". In: Proceedings of the AAAI Conference on Artificial Intelligence 35.6 (May 2021), pp. 5303–5311.
- Yae Jee Cho, Divyansh Jhunjhunwala, Tian Li, Virginia Smith, and Gauri Joshi. “To Federate or Not To Federate: Incentivizing Client Participation in Federated Learning”. In: Workshop on Federated Learning: Recent Advances and New Challenges (in Conjunction with NeurIPS 2022). 2022.

**Questions:**

1. Could you explain what is the technical novelty of the paper, and why Theorem 1. is not an immediate corollary from the analysis of Cho et al. (2022)?
2. Why is the convergence result stated in terms of $F$. If the purpose is to eliminate bad clients, then it would make more sense to report the convergence results in terms of $F_g$.
3. Please precise what is the evaluation metric; do you report the average/weighted average test loss/accuracy across all/good clients?

---

### Official Review · Reviewer_3krf · 2023-11-02

**Soundness:** 2 fair
**Presentation:** 2 fair
**Contribution:** 2 fair
**Rating:** 3
**Confidence:** 4

**Summary:**

The paper proposes FedSRC, a new federated learning algorithm with adaptive client selection. The paper finds that bad clients tend to have higher inference loss with the global model. Thus, the server first collects the inference loss and uses clustering method to determine a threshold. Then, the clients with inference loss below the threshold continue the training for the current round and the other clients stop training to save computation costs. Experiments show that FedSRC achieves lower test loss than the other baselines.

**Strengths:**

1. The paper provides a theoretical convergence analysis of the proposed algorithm.

2. The proposed algorithm is simple and easy to understand.

3. FedSRC achieves better performance than the other baselines.

**Weaknesses:**

1. The idea is not new. Client selection based on the inference loss has been applied in existing studies, e.g., oort [1]. The proposed approach is very similar to oort.

[1] Oort: Efficient Federated Learning via Guided Participant Selection

2. The motivation is not clear. The paper claims that existing client selection leaks the client IDs. However, the client IDs are still leaked when aggregating the local models as the local data size is required. A clear explanation of why the client IDs should be private and how the proposed approach differs from existing client selection approaches is required.

3. Experiments are not solid. The FL algorithms for client selection (e.g., oort [1]) are not compared. Also, it’d be better to add experiments on scalability and different non-IID degrees.

**Questions:**

1. Why the client IDs are private information? Why FedSRC can protect the client IDs? The models are still sent for weighted aggregation like FedAvg. The server can infer the client ID by the data size. Moreover, the server usually knows who is the sender in a communication protocol.

2. Can you add experiments to compare the FL baselines with client selection?

3. What is the number of clients in the experiments? Can you add experiments by varying the number of clients and the non-IID degree?

---

### Official Review · Reviewer_yhN4 · 2023-11-02

**Soundness:** 2 fair
**Presentation:** 3 good
**Contribution:** 2 fair
**Rating:** 3
**Confidence:** 4

**Summary:**

This paper proposed a novel algorithm, namely FedSRC, for handling data corruption from the client side to enhance the efficiency of federated learning. A client in FedSRC needs to download the model to check its local test loss, regardless of its participation. The convergence analysis is provided in this paper. The experimental results also show the effectiveness of FedSRC.

**Strengths:**

1. This paper attempts to regulate client participation from the client side. A client in the proposed algorithm will decide whether it will participate in the next communication round by checking its local test loss on the global model.
2. The convergence analysis is provided in this paper. The convergence bound can reveal the effect of the client selection strategy.
3. The experimental results also show the effectiveness of FedSRC.

**Weaknesses:**

1. There are a lot of studies on the theoretical analysis of FL convergence under clients with data quality issues, e.g., the Non-IID setting. What is the key contribution of this theoretical analysis?
2. In the proposed algorithm, a client stops training and drops from the FL round if its test loss exceeds the participation threshold. However, the previous work aimed to schedule the client with high loss values [1].
3. The contributions of this paper are not very strong. The proposed algorithm needs a two-round communication to achieve a global model.
4. The benchmark algorithms used in this paper are too old. In addition, these benchmark algorithms were designed for malicious clients instead of data quality issues.

The reference mentioned is shown as follows:
[1] Y. J. Cho, J. Wang, and G. Joshi, “Client selection in federated learning: Convergence analysis and power-of-choice selection strategies,” in AISTATS, Mar. 2022.

**Questions:**

1. Please highlight your key contribution to the theoretical analysis on convergence bound. There are a lot of studies on the theoretical analysis of FL convergence under clients with data quality issues, e.g., the Non-IID setting.  The authors should explain the difference between the analysis results and the previous works.
2. In the proposed algorithm, a client stops training and drops from the FL round if its test loss exceeds the participation threshold. However, the previous work aimed to schedule the client with high loss values [1].
3. The contributions of this paper are not very strong. In addition, the proposed algorithm will add some extra communication costs, because it needs a two-round communication to achieve a global model.
4. The benchmark algorithms used in this paper are too old. In addition, these benchmark algorithms selected are not appropriate, because they were designed for malicious clients instead of data quality issues.

The reference mentioned is shown as follows:
[1] Y. J. Cho, J. Wang, and G. Joshi, “Client selection in federated learning: Convergence analysis and power-of-choice selection strategies,” in AISTATS, Mar. 2022.

---

### Official Review · Reviewer_2b2m · 2023-11-03

**Soundness:** 3 good
**Presentation:** 3 good
**Contribution:** 2 fair
**Rating:** 5
**Confidence:** 4

**Summary:**

The paper introduces FedSRC (Federated Learning with Self-Regulating Clients), a novel approach to address slow convergence and suboptimal performance in Federated Learning (FL) due to data heterogeneity and quality concerns. FedSRC empowers clients to autonomously regulate their participation by using a lightweight "checkpoint" to evaluate their training's impact on the global model. This novel approach significantly reduces communication and computational overhead while improving FL performance. The paper also provides a rigorous theoretical analysis of FL convergence under data quality issues, showcasing the effectiveness of FedSRC. Extensive evaluations on diverse datasets validate the efficacy of FedSRC, notably reducing communication costs by up to 30% and computational costs by up to 55% over equivalent rounds.

**Strengths:**

1. Originality: FedSRC introduces an innovative, self-regulating client mechanism, departing from conventional strategies and reducing communication and computational overhead.

2. Quality: The paper maintains high-quality standards through its structured elucidation of FedSRC, theoretical analysis, and comprehensive empirical evaluations across diverse datasets.

3. Clarity: The paper excels in lucidly conveying core concepts, methodologies, and findings in a coherent and accessible manner, ensuring broad readership comprehension.

**Weaknesses:**

1. Baseline Choice Concerns: The reliance on honest client reporting may lack robustness against malicious clients in contrast to Byzantine fault-tolerant baseline methods.

2. Data Quality and Non-IID Issues: The inference loss-based checkpoint might struggle to discern between unreliable data and non-IID aspects, potentially leading to deteriorating model performance.

3. Lack of Replicability and Experimentation: Explicit details regarding data partitioning methods and other parameters are absent, hindering replicability. Additionally, the paper lacks experimentation with various non-IID partitioning strategies and different data corruption ratios, which could illuminate the algorithm's robustness.

**Questions:**

1.	The paper could benefit from a more extensive comparative analysis of its performance with a broader range of Federated Learning baselines, particularly those that also need an honest and trustworthy client environment. Meanwhile, it is advisable to provide a detailed description of the experimental setups and hyperparameters for these baseline methods to allow for a more comprehensive evaluation.

2.	Additionally, the paper should consider conducting comparative experiments under various non-IID data partitioning strategies and explore the distribution of data possessed by clients that were discarded. This is particularly important due to concerns regarding the algorithm's performance in non-IID scenarios, as it appears to have a propensity for outright discarding all extremely non-IID clients.  Examining a variety of non-IID scenarios would provide a more realistic assessment of its capabilities.

3.	Furthermore, the paper should clarify the protocol for client re-inclusion. If a client's Inference Loss falls below the specified threshold in one round, will it automatically rejoin the subsequent training rounds? This aspect requires attention since repeated loss checks could lead to increased communication and computational overhead. It is essential to establish a mechanism to determine when clients with Inference Loss below the threshold can re-enter the FL process, preventing the permanent exclusion of potentially valuable clients.

---

### Meta-Review · Area_Chair_RmBp · 2023-12-07

**Metareview:**

This paper proposes a new adaptive client selection technique for federated learning. The theoretical analysis is also established to demonstrate the effectiveness of the client selection strategy. Preliminary experiments demonstrate the efficacy of the proposed approach.

Strengths:

(1)   This paper is well written.

(2)   The client selection is very important for federated learning. The motivation for this work is clear.

(3)   The authors also try to provide a theoretical analysis of the proposed methods.

Weaknesses:

(1)   Although the authors provide the theoretical convergence for the proposed method. However, the technical challenges and technical contributions for analyzing the convergence are not clear.

(2)   The proposed approach requires two-round communications to achieve a global model, which limits the practical application of the proposed method.

(3)   The compared baselines are too old to demonstrate the efficacy of the proposed method.

In addition, the authors don’t provide any response, the raised concerns by the reviewers still exist. Therefore, I recommend rejection.

**Justification For Why Not Higher Score:**

N/A

**Justification For Why Not Lower Score:**

N/A

---

### Decision · Program_Chairs · 2024-01-16

Reject